# Advances in Virtual Cutting Guide and Stereotactic Navigation for Complex Tumor Resections of the Sacrum and Pelvis: Case Series with Short-Term Follow-Up

**DOI:** 10.3390/bioengineering10121342

**Published:** 2023-11-22

**Authors:** Takashi Hirase, Grant R. McChesney, Lawrence Garvin, Karthik Tappa, Robert L. Satcher, Alexander F. Mericli, Laurence D. Rhines, Justin E. Bird

**Affiliations:** 1Department of Orthopedic Oncology, Division of Surgery, The University of Texas MD Anderson Cancer Center, 1515 Holcombe Blvd, Houston, TX 77030, USA; tak.hirase@gmail.com (T.H.); lawrence.garvin1906@gmail.com (G.L.II);; 2Department of Breast Imaging, Division of Diagnostic Imaging, The University of Texas MD Anderson Cancer Center, 1515 Holcombe Blvd, Houston, TX 77030, USA; kktappa@mdanderson.org; 3Department of Plastic Surgery, Division of Surgery, The University of Texas MD Anderson Cancer Center, 1515 Holcombe Blvd, Houston, TX 77030, USA; afmericli@mdanderson.org; 4Department of Neurosurgery, Division of Surgery, The University of Texas MD Anderson Cancer Center, 1515 Holcombe Blvd, Houston, TX 77030, USA

**Keywords:** virtual cutting guide, computer-assisted surgical planning, image-guided navigation, sacropelvic tumor, sacropelvic reconstruction, osteotomy, en bloc resection

## Abstract

Primary malignancies of the sacrum and pelvis are aggressive in nature, and achieving negative margins is essential for preventing recurrence and improving survival after en bloc resections. However, these are particularly challenging interventions due to the complex anatomy and proximity to vital structures. Using virtual cutting guides to perform navigated osteotomies may be a reliable method for safely obtaining negative margins in complex tumor resections of the sacrum and pelvis. This study details the technique and presents short-term outcomes. Patients who underwent an en bloc tumor resection of the sacrum and/or pelvis using virtual cutting guides with a minimum follow-up of two years were retrospectively analyzed and included in this study. Preoperative computer-assisted design (CAD) was used to design osteotomies in each case. Segmentation, delineating the tumor from normal tissue, was performed by the senior author using preoperative CT scans and MRI. Working with a team of biomedical engineers, virtual surgical planning was performed to create osteotomy lines on the preoperative CT and overlaid onto the intraoperative CT. The pre-planned osteotomy lines were visualized as “virtual cutting guides” providing real-time stereotactic navigation. A precision ultrasound-powered cutting tool was then integrated into the navigation system and used to perform the osteotomies in each case. Six patients (mean age 52.2 ± 17.7 years, 2 males, 4 females) were included in this study. Negative margins were achieved in all patients with no intraoperative complications. Mean follow-up was 38.0 ± 6.5 months (range, 24.8–42.2). Mean operative time was 1229 min (range, 522–2063). Mean length of stay (LOS) was 18.7 ± 14.5 days. There were no cases of 30-day readmissions, 30-day reoperations, or 2-year mortality. One patient was complicated by flap necrosis, which was successfully treated with irrigation and debridement and primary closure. One patient had local tumor recurrence at final follow-up and two patients are currently undergoing treatment for metastatic disease. Using virtual cutting guides to perform navigated osteotomies is a safe technique that can facilitate complex tumor resections of the sacrum and pelvis.

## 1. Introduction

Malignancies originating in the sacrum and pelvis are exceedingly rare clinical conditions that pose an immense surgical challenge. Irrespective of the specific type of malignancy involved, these tumors are known for their aggressive nature. En bloc resection, a surgical approach entailing the removal of both the entirety of the tumor and a portion of surrounding healthy tissue, often emerges as the sole intervention that has demonstrated a definitive survival benefit [1,2]. It is well documented that when resections are inadequate, the risk of tumor recurrence is alarmingly high, reaching up to 100% in cases of intralesional resections and 70% in cases of marginal resections [3,4]. The complexity of the anatomical structures in the sacrum and pelvis, coupled with their proximity to vital neighboring structures, poses a significant challenge for orthopedic and neurosurgical oncologists when attempting to achieve optimal tumor resection and subsequent reconstruction. Additionally, the risk of iatrogenic intraoperative vascular and visceral injuries hovers between 20% and 33% during attempts to achieve the required margins [5,6].

In this context, the utilization of image-guided navigation emerges as a valuable tool for enhancing precision in the quest to achieve negative surgical margins while simultaneously mitigating potential complications in sacral and pelvic tumor resections [7,8,9,10,11,12]. Over the past two decades, the field of computer-assisted design (CAD) has witnessed a surge in popularity, particularly in the realm of maxillofacial surgery, where its application has been instrumental in refining surgical precision [13,14,15]. Currently, CAD is considered as a valuable tool that enhances various aspects of the surgical process, including preoperative planning, customization and personalization of patient-specific models, simulation and training, implant design and fabrication, communication and collaboration among multidisciplinary departments and intraoperative guidance. Additionally, 3D CAD models can help potentially reducing surgery time, contributing to cost saving and in-turn simulation of outcomes [16]. Notably, the fusion of CAD templates with image-guided navigation has been recognized as a dependable method for executing intricate facial reconstructions within maxillofacial surgery [17]. Similarly, within the subspecialties of spine and orthopedic surgery, the image-guided bone scalpel, which is a precision ultrasound-powered bone-cutting tool, has recently emerged as an adjunct that enables precise tumor resection [18].

Traditionally, CAD is used to create 3D-printed physical templates and cutting guides to assist the surgeon with osteotomies, which are bone cuts necessary to execute the surgery. While useful for maxillofacial surgeries, the senior authors have found certain drawbacks to this technology. These physical cutting guides are often characterized as bulky, unwieldy and challenging to use within the highly intricate pelvic space. Additionally, there is always additional labor, time and expense involved in modeling the custom cutting guides and 3D printing them. These limitation spurred an innovative solution which, to the best of our knowledge, has not been previously described: the combination of CAD-created STL files and the use of the image-guided bone scalpel to create what we have termed, “virtual cutting guides” for pelvic osteotomies.

This series represents our initial experience with utilizing virtual cutting guides by overlaying the CAD templates with image-guided navigation and performing navigated osteotomies with an image-guided bone scalpel. It is imperative to note that such techniques have not been previously described for complex tumor resections within the sacrum and pelvis. This manuscript seeks to present the outcomes derived from six cases in which these innovative techniques were applied, and furthermore, it offers a comprehensive review of the existing literature pertaining to similar methods used in the past.

## 2. Methods

At a single institution between January 2017 and August 2019, a retrospective analysis was performed of patients who underwent an en bloc tumor resection and reconstruction of the sacrum and/or pelvis using virtual cutting guides and navigated osteotomy. STROBE (STrengthening the Reporting of OBservational studies in Epidemiology) guidelines were implemented [19]. Inclusion criteria were any patients aged 18 or above with a minimum follow-up of 24 months. Exclusion criteria were patients lacking a preoperative CAD, metastatic disease, and revision surgery. Demographic data and outcome measures were obtained retrospectively through electronic medical records. Obtained demographics included age, gender, American Anesthesiologists’ Society (ASA) class, BMI, comorbidities, diagnosis for surgery including histologic grade and Musculoskeletal Tumor Society (MSTS) stage, and follow-up. Obtained outcome measures included surgical margins, operative time, estimated blood loss (EBL), perioperative complications, tumor recurrence, 30-day readmission rates, 30-day reoperation rates, 1-year mortality rates, discharge disposition, and postoperative length of stay (LOS). Postoperative LOS was defined as the number of inpatient nights from surgery (or the last surgery if a staged procedure) to time of discharge to their home or a rehabilitation facility.

### 2.1. CAD Development

Prior to surgery, a fine-cut (0.5 mm slice thickness) computed tomography (CT) scan of the involved region is performed. CT scans of the unilateral or bilateral fibulas are also obtained if reconstruction with free vascularized fibular grafts (FVFG) is planned. Segmentation, delineating the tumor from normal tissue, was performed by the senior author using preoperative CT scans and MRI. Different imaging modalities are first registered together to aid in the anatomy’s segmentation. Bony tissues as well as calcified tumor were segmented from the CT images, and muscular anatomies as well as cartilaginous tumor were segmented from the MR images. Working with a team of biomedical engineers, virtual surgical planning was performed to create osteotomy lines on the preoperative CT. These segmented anatomies as well as virtual cutting planes were saved as objects in an STL file format. The STL files were then uploaded into the stereotactic navigation system, aligned and overlaid on to the intraoperative CT. These images are transferred into Materialise Mimics Medical Innovation Suite V 25.0 software (Materialise/DePuy Synthes, Leuven, Belgium). CAD development is performed through a multidisciplinary collaboration between the surgeons and engineers. Using this software, tumor and surrounding anatomies are segmented and reviewed by the radiologist. Once the required anatomies are segmented, 3D volumes of these anatomies are rendered, and respective CAD models are generated. Virtual 3D cutting planes are then placed on these CAD models at areas of resection (Figure 1).

If sacropelvic reconstruction with FVFG is planned, the appropriate resection planes of the fibula are established and incorporated into the post-resection reconstruction model (Figure 2). CAD models of tumor, surrounding anatomies and cutting planes are then exported as STL files. While exporting these as STL files, fiducial markings are placed on every STL file to help with location and realignment.

These STL files along with the preoperative DICOMs used for segmentation are loaded on to the BrainLab Origin Server 3.2 and opened in the iPlan Spine 3.0.6 app. We have noticed that importing these STL files into the software disorients and misaligns objects in the virtual space. Using Advanced Object Planning tab in iPlan and prior placed fiducial markings, STL anatomies and virtual cutting guides are reoriented, aligned and saved. This file is loaded on to the BrainLab Curve (not Kolibri) and opened in Spine and Trauma 3D 2.6.0. Once the intraoperative scan is completed, the preoperative CT and intraoperative CT are aligned using the ImageFusion 4.0 app in the BrainLab Curve. All the anatomies and individual virtual guides can be turned on or off individually as needed by the surgeon.

### 2.2. Intraoperative CAD Utilization

The steps for intraoperative CAD utilization are highlighted in Figure 3. The navigation array is first applied at a location away from the planned osteotomy sites (Figure 3B). Intraoperative CT images are then obtained and imported into the BrainLab Kolibri (BrainLab, Munich, Germany) navigation system. The CAD is overlaid on to the CT images to create a real-time, virtual 3D reconstruction image of the tumor and the planned osteotomy planes. Accuracy of the scan and the CAD overlay is confirmed using the navigated probe on multiple visibly identifiable anatomical landmarks in the coronal, sagittal, and axial planes. A tracking probe clamp is attached to the BoneScalpel (Misonix, Farmingdale, NY, USA) and registered to the navigation platform (Figure 3C). The navigated BoneScalpel is then used to execute the planned osteotomies while ensuring that all bone cuts are made precisely along the osteotomy planes depicted on the virtual guide (Figure 3D).

## 3. Results

Six patients (mean age 52.2 ± 17.7 years, 2 males, 4 females) underwent CAD-merged image-guided operations during the study period (Table 1). Mean follow-up was 38.0 ± 6.5 months (range, 24.8–42.2). Staged procedures were performed in three patients. Apart from their initial cancer diagnosis, there were few underlying health conditions, with three patients having no comorbidities, two patients having asthma, and one patient having a history of smoking. One patient had Musculoskeletal Tumor Society (MSTS) stage IIB radiation-induced osteosarcoma involving the sacrum and right ilium. One patient had stage IIB primary osteosarcoma involving the sacrum and left ilium. Three patients had stage IIB primary chondrosarcoma involving the pelvis and/or sacrum. One patient had stage III secondary chondrosarcoma with a prior history of multiple hereditary enchondromatosis (MHE). One patient with a primary chondrosarcoma of the acetabulum underwent a pelvis and hip joint reconstruction using a custom endoprosthetic implant. All cases were diagnosed preoperatively with core needle biopsy and margins were confirmed with intraoperative frozen specimens.

Negative margins were achieved in all patients with no intraoperative complications. There were no technical difficulties involving the described technology when carrying out the surgery. Mean operative time was 1229 min (range, 522–2063). Mean EBL was 4425 mL (range, 300–12,150). Mean length of stay (LOS) was 18.7 days (range 7–47). All patients were discharged to a rehabilitation facility. There were no 30-day readmissions, 30-day reoperations, or 2-year mortality at final follow-up. One patient was complicated by flap necrosis, which was successfully treated with debridement and primary closure. One patient had local tumor recurrence at final follow-up. One patient with an initial diagnosis of stage IIB radiation-induced osteosarcoma had local recurrence at 14-month follow-up and is currently undergoing treatment for metastatic disease. One patient with MHE and an initial diagnosis of stage III secondary chondrosarcoma subsequently underwent marginal resections of metastatic disease to the left groin and left shoulder at 24 months and 42 months, respectively, after initial surgery.

## 4. Discussion

This study describes the surgical technique consisting of using virtual cutting guides to perform navigated osteotomies to safely and accurately perform complex tumor resections of the sacrum and pelvis. Although previous studies report the value of CAD and image-guided navigation, to our knowledge, no prior studies have reported the successful use of merging the two techniques to develop virtual cutting guides, a form of augmented-reality-assisted surgery.

To minimize the incidence of recurrences, an en bloc resection with wide margins is the preferred surgical treatment for primary osteosarcoma and high-grade chondrosarcoma [20,21]. However, performing wide resections in anatomically complex regions such as the lumbar spine, sacrum, and pelvis present challenges for surgeons to obtain adequate margins whilst minimizing morbidity, increasing the risk for intralesional resection and local recurrences [22]. A study by Jeys et al. reported a series of 539 patients undergoing wide resection of primary bone tumors of the sacrum or pelvis using standard surgical techniques without the use of CAD or navigation and found a 29% intralesional resection rate and a 27% local recurrence rate. This was compared to a series of 31 patients who underwent the same procedure using image-guided navigation and reported an 8.7% intralesional resection rate and a 13% local recurrence rate [11]. The authors speculated that although there were significant reductions in intralesional resection and recurrence rates with the use of navigation, using a non-navigated osteotome to perform the osteotomies may reduce the accuracy of the resection planes. Furthermore, without the use of CAD to be overlaid with the navigation imaging, the authors are unable to perform osteotomies precisely through planned resection planes. A recent study by Towner et al. introduced a technique in which an ultrasonic bone scalpel was registered with image-guided navigation to perform precise osteotomies under navigation for spine tumor resections [18]. The authors demonstrated the safety of its use in spine tumor resection and introduced potential advantages including higher precision and decreased operative times. CAD techniques have been recently introduced and are gaining popularity for improving precision in the field of maxillofacial surgery [13,14,15]. A case report by Nguyen et al. reported the first successful application of merging CAD with image-guided navigation for maxillofacial reconstruction of complex facial fractures [17]. Our study combined the use of these novel techniques by developing virtual cutting guides and performing navigated osteotomies to successfully obtain negative margins with no local recurrence in 100% of patients within our series at 12-month follow-up.

In addition to aiding in obtaining negative margins, CAD can effectively be utilized for sacropelvic reconstructions using FVFG as well as for custom endoprosthesis implantation for pelvis and hip reconstruction [23]. Sacropelvic tumor resections often result in large osseous defects and spinopelvic discontinuity and instability. Such resections would benefit from the use of FVFG to reconstruct the sacropelvic junction [23]. However, using the appropriately sized grafts are essential for obtaining union and stability after reconstruction [23,24]. The use of CAD allows the surgeon to preselect the exact length, orientation, and the number of grafts required for appropriate reconstruction, thus maximizing the biomechanical construct and possibly enhancing union rates [23]. Within our case series of six patients, three patients underwent sacropelvic reconstruction with FVFG and the grafts were obtained and appropriately sized using CAD and image-guided navigation (Figure 4).

All three patients achieved bony union at 12-month follow-up. With regard to the use of CAD for custom endoprosthesis implantation for pelvis and hip reconstruction, precise osteotomies are necessary to successfully implant a custom endoprosthesis. Recent literature has demonstrated significant advances in 3D technology for this purpose by developing 3D models of the patient anatomy and the custom endoprosthesis as well as 3D cutting guides to be used intraoperatively [25,26]. However, the intraoperative use of 3D models and cutting guides are often challenging due to presence of soft tissues and vital structures surrounding the planned osteotomy sites. Thus, incorporating CAD and virtual gutting guides for executing precise osteotomies is not only useful for en bloc tumor resections but also particularly helpful in preparing the hip and pelvis for a custom endoprosthesis. With our case series, one patient with a primary chondrosarcoma of the acetabulum successfully underwent a pelvis and hip joint reconstruction using a custom endoprosthetic implant without complications (Figure 5).

There are limitations with the use of CAD and image-guided navigation techniques. CAD is developed using CT images acquired preoperatively, which is overlaid with CT images obtained intraoperatively for image-guided navigation. Thus, particularly with fast-growing tumors, there may be slight anatomical differences seen after the image merging process. Therefore, we suggest obtaining the preoperative CT for CAD development as close as possible to the day of surgery. Furthermore, prior studies have described image-to-patient registration errors of up to 2 mm [27,28,29]. Within our study, the accuracy of the scans and the CAD overlays were confirmed using the navigated probe on multiple visibly identifiable anatomical landmarks in the coronal, sagittal, and axial planes, and a less than 1 mm. registration error was confirmed prior to proceeding with the planned osteotomies. Finally, segmentation of tumor can be performed on MRI and merged with intraoperative CT; however, this is often suboptimal due to changes in patient positioning between the two scans.

There are several limitations to this study. Notably, the retrospective nature of the case series with a small sample size is vulnerable to selection bias and is insufficient in terms of statistical power for conducting an analysis. The lack of a control group or comparative analysis complicates the determination of the precise benefits of this method in comparison to traditional approaches. Furthermore, although all reported cases utilized the method of merging CAD with image-guided navigation, the complexity of each case was variable in nature and the aggregate result may not be reflective of all outcomes. This study was also limited by a minimum follow-up period of 24 months; thus, we are unable to comment on survival advantages beyond this time using this operative technique. Future studies should include prospective comparative studies with a larger sample size and longer follow-up that allows for further determination of the efficacy of this surgical technique. Additionally, future work that also applies this technology within other areas of the spine would also contribute to enhancing our comprehension of the effectiveness of this approach.

## 5. Conclusions

In conclusion, our study demonstrates a novel method of utilizing virtual cutting guides as a safe technique to perform navigated osteotomies that can facilitate complex tumor resections of the sacrum and pelvis. The CAD models, generated from the preoperative images, can be obtained within the institution in timely fashion as well as cost-effectively and could be successfully laid over the intraoperative images to provide guidance during the surgical procedure.

## Figures and Tables

**Figure 1 bioengineering-10-01342-f001:**
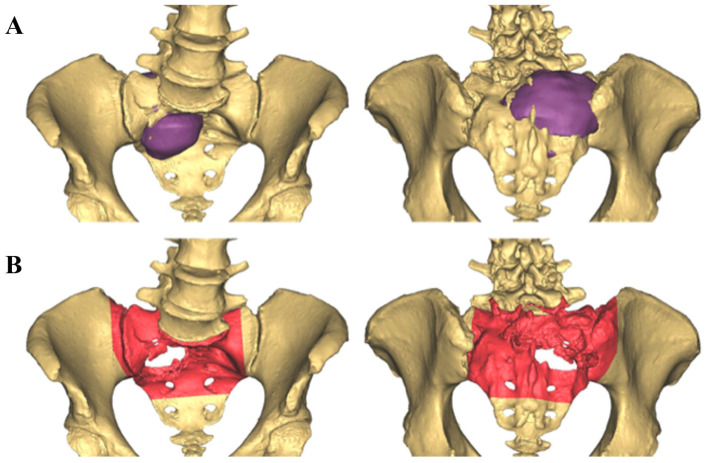
(**A**) Tumor involvement within the preoperative 3D CAD. The purple area represents the tumor. (**B**) Planned resection planes for en bloc resection as represented in red.

**Figure 2 bioengineering-10-01342-f002:**
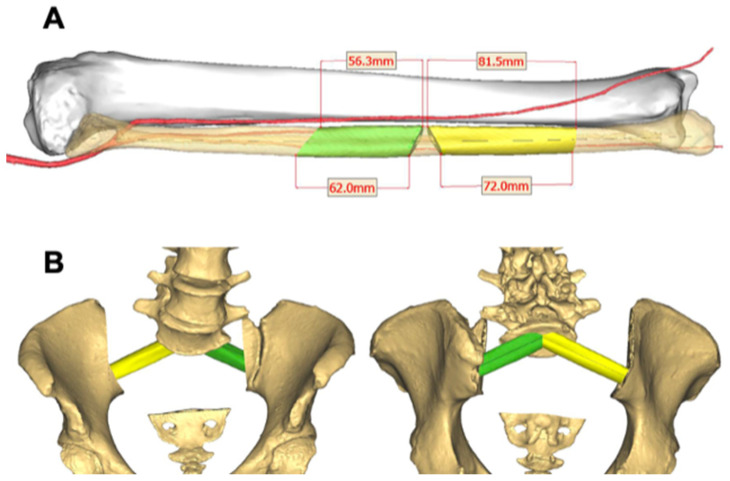
(**A**) Resection planes of the vascularized fibula grafts represented in green and yellow. (**B**) Virtual sacropelvic reconstruction model with the templated fibula grafts providing a perfect fit within the resected voids.

**Figure 3 bioengineering-10-01342-f003:**
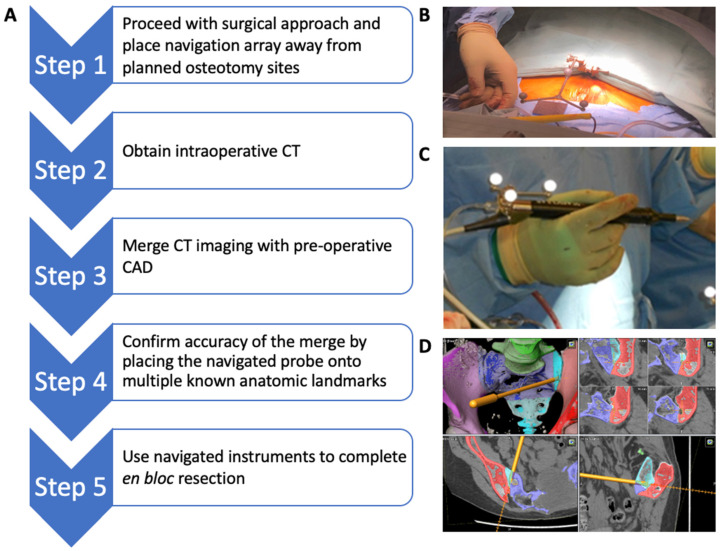
(**A**) Outline for intraoperative CAD utilization. (**B**) Intraoperative placement of navigation array. (**C**) Attachment of tracking probe clamp to the BoneScalpel. (**D**) Virtual cutting guide and anterior osteotomies performed using navigated BoneScalpel.

**Figure 4 bioengineering-10-01342-f004:**
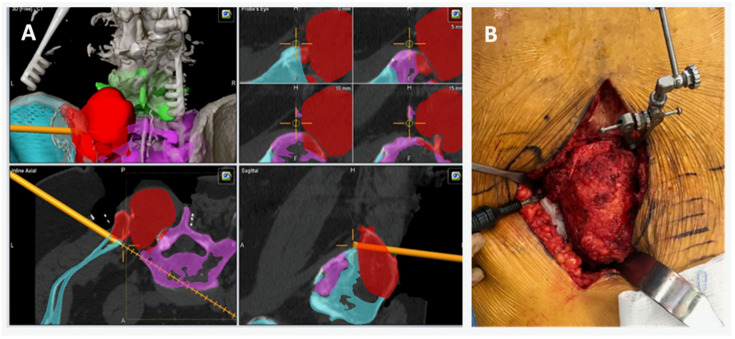
(**A**) Virtual cutting guide with virtual representations of the navigated BoneScalpel for performing posterior osteotomies around the tumor (red). (**B**) Intraoperative photo demonstrating navigated BoneScalpel. The use of the navigated bone scalpel with virtual cutting guides significantly decreased the size of the incision and degree of required soft tissue exposure.

**Figure 5 bioengineering-10-01342-f005:**
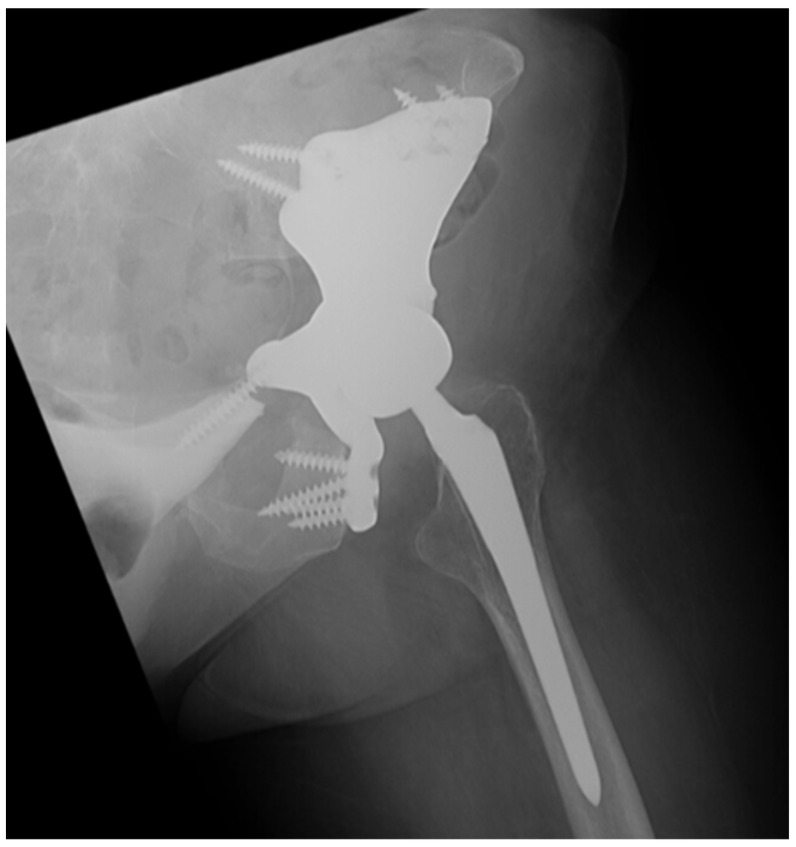
Left hip radiograph of a patient with primary chondrosarcoma of the acetabulum that underwent a pelvis and hip joint reconstruction using a custom endoprosthetic implant.

**Table 1 bioengineering-10-01342-t001:** Patient Demographics and Outcomes.

Case No.	1	2	3	4	5	6
Age (yrs), Sex	53, F	25, F	60, F	39, F	61, M	75, M
ASA Class	3	3	3	3	3	3
BMI (kg/m^2^)	17.5	18.0	25.9	31.4	28.6	30.4
Follow-Up (months)	24.8	39.7	40.3	42.2	41.5	39.2
Comorbidities	None	None	None	Asthma	Former smoker	Asthma
Diagnosis	Radiation-Induced Osteosarcoma	Primary Osteosarcoma	Primary Chondrosarcoma	MHE, Secondary Chondrosarcoma	Primary Chondrosarcoma	Primary Chondrosarcoma
Histologic Grade	High grade	High grade	High grade	De-differentiated	High grade	High grade
MSTS Stage	IIB	III	IIB	III	IIB	IIB
Location	Sacrum, Rt ilium	LS spine, Lt ilium	Lt acetabulum	Lt ilium	Sacrum, Lt ilium	Sacrum, Lt ilium
Procedures Performed	Stage 1: L5-S1 anterior discectomy, anterior sacral osteotomy, fibula flap harvest Stage 2: sacrectomy, L3-pelvis PSIF, L5-pelvis ASF with vascularized fibula flap, VRAM flap	Stage 1: Rt T12-pelvis PSIF, L3-5 laminectomy, fibula flap harvest; Stage 2: L4, L5 vertebrectomy, sacrectomy, Lt type I internal hemipelvectomy, Lt L3-pelvis PSIF Stage 3: L3-pelvis ASF with vascularized fibula flap	Lt type I-II internal hemipelvectomy, custom endoprosthetic pelvic and hip joint reconstruction	Lt Type I internal hemipelvectomy, vascularized fibula autograft reconstruction	Stage 1: L3-S3 laminectomy, L5 vertebrectomy, L3-pelvis PSIF; Stage 2: Lt type I internal hemipelvectomy, Lt partial sacrectomy, L5-pelvis ASF with vascularized fibula autograft	L5-S1 laminectomy, partial sacrectomy, Lt ilium osteotomy, pedicled Rt gluteus maximus flap
EBL (mL)	12,150	5500	1400	400	6800	300
OR Time (min)	2063	1739	522	543	1994	515
Margins	Negative	Negative	Negative	Negative	Negative	Negative
Intraoperative Complications	None	None	None	None	None	None
LOS (days)	19	47	16	12	11	7
Discharge Disposition	Rehab facility	Rehab facility	Rehab facility	Rehab facility	Rehab facility	Rehab facility
30-Day Readmission	No	No	No	No	No	No
30-Day Reoperation	No	No	No	No	No	No
2-Year Mortality	No	No	No	No	No	No
Postoperative Complications	None	Flap necrosis	None	None	None	None
Local Recurrence	Yes	No	No	No	No	No
Metastatic Disease	Yes	No	No	Yes, left groin, left shoulder s/p resection	No	No

ASA, American Society of Anesthesiologists; BMI, body mass index; MSTS, Musculoskeletal Tumor Society; EBL, estimated blood loss; OR, operating room; LOS, length of stay; F, female; M, male; LS, lumbosacral; Lt, left; Rt, right; PSIF, posterior spinal instrumented fusion; ASF, anterior spinal fusion; VRAM, vertical rectus abdominis musculocutaneous; MHE, multiple hereditary enchondromatosis.

## Data Availability

The datasets generated and/or analyzed during the current study are available from the corresponding author upon request.

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
