# Peer review of "Advances in Virtual Cutting Guide and Stereotactic Navigation for Complex Tumor Resections of the Sacrum and Pelvis: Case Series with Short-Term Follow-Up"

_bioengineering, 2023, doi:10.3390/bioengineering10121342_

Round 1

Reviewer 1 Report

Comments and Suggestions for Authors

The study explores the innovative use of virtual cutting guides and stereotactic navigation for safe and reliable complex tumor resections in the sacrum and pelvis. The authors state that this approach achieved 100% negative margins and no local recurrence in a case series of six patients.

The proposed method is interesting, but the entire structure of the article needs to be revised.

The abstract is too long (more than 400 words). The main methods should be described more concisely, and the main results and conclusions should be summarized.

The introduction is too brief (less than 400 words) and is not easy to understand for scientists outside the particular field of research. It should contextualize and justify the study's purpose and significance, discuss the current state of the research field, and highlight the main contributions of the research. The introduction paragraph should also be numbered as "1."

The methods section should be described in more detail. The results section is also too concise. It should offer a succinct and accurate account of the experimental findings, their interpretation, and the resulting experimental implications.

The figures should be described more accurately.

In my opinion, some of the content in the discussion section would be more relevant in the introduction paragraph. The authors should contextualize the results by relating them to prior studies and hypotheses, and discuss their broader implications. Future work could be highlighted in the discussion or conclusions section.

Author Response

The study explores the innovative use of virtual cutting guides and stereotactic navigation for safe and reliable complex tumor resections in the sacrum and pelvis. The authors state that this approach achieved 100% negative margins and no local recurrence in a case series of six patients. The proposed method is interesting, but the entire structure of the article needs to be revised.

The abstract is too long (more than 400 words). The main methods should be described more concisely, and the main results and conclusions should be summarized.

Response: The authors appreciate your time for reviewing the manuscript. We have now addressed this and decreased the abstract word count to 333.

The introduction is too brief (less than 400 words) and is not easy to understand for scientists outside the particular field of research. It should contextualize and justify the study's purpose and significance, discuss the current state of the research field, and highlight the main contributions of the research. The introduction paragraph should also be numbered as "1."

Response: We agree with this assessment. We have now rephrased and expanded on the introduction section to 500 words. We have also changed the section number to ‘1’.

The methods section should be described in more detail. The results section is also too concise. It should offer a succinct and accurate account of the experimental findings, their interpretation, and the resulting experimental implications.

Response: We agree with this assessment. We have now expanded the methods and results section as highlighted. The method section also contains two sub-sections describing in detailed procedure of converting a DICOM file to STL file and then process of using these STL files in navigation system during the procedure.

The figures should be described more accurately.

Response: The authors agree with this assessment. We have now addressed this as highlighted within the figures.

In my opinion, some of the content in the discussion section would be more relevant in the introduction paragraph. The authors should contextualize the results by relating them to prior studies and hypotheses, and discuss their broader implications. Future work could be highlighted in the discussion or conclusions section.

Response: The authors agree with this assessment. We have added some content in the discussion section within the introduction section to enhance better understanding of the background content. We have also now added additional statements regarding future work within last paragraph of the discussion section as highlighted.

Reviewer 2 Report

Comments and Suggestions for Authors

Very informative and very innovative case series on a rare but relevant condition providing "Advances in Virtual Cutting Guide and Stereotactic Navigation for Complex Tumor Resections of the Sacrum and Pelvis. 

The authors should emphasize the retrospective nature and short-term follow up of their study.

The authors should adhere to the STROBE criteria for retrospective observational data.

Comments on the Quality of English Language

The manuscript would profit from Native english language editing.

Author Response

Very informative and very innovative case series on a rare but relevant condition providing "Advances in Virtual Cutting Guide and Stereotactic Navigation for Complex Tumor Resections of the Sacrum and Pelvis. 

The authors should emphasize the retrospective nature and short-term follow up of their study.

Response: The authors appreciate your time for reviewing the manuscript. We have now addressed this as highlighted within last paragraph of the discussion section.

The authors should adhere to the STROBE criteria for retrospective observational data.

Response: The authors have adhered to the STROBE criteria and has now included this within the methods section and added the citation [21].

Reviewer 3 Report

Comments and Suggestions for Authors

This study presents a promising technique for challenging sacral and pelvic tumor resections using virtual cutting guides. The use of preoperative CAD, real-time stereotactic navigation, and ultrasound-powered tools demonstrates innovation in improving surgical precision. Some concerns:

1. The author's article exhibits concerns with respect to citation practices. The background section, in particular, demonstrates an overabundance of citations. Conversely, within the main body of the text, there are instances where critical points, warranting appropriate citation, lack the requisite referencing.

2. The distinctions in color within Figure 1 require clarification and elucidation.

3.The limited sample size in the experiment raises questions regarding the overall validity and significance of the research findings.

4. The absence of a control group or comparative analysis makes it challenging to ascertain the specific advantages of this method over conventional approaches.

Author Response

This study presents a promising technique for challenging sacral and pelvic tumor resections using virtual cutting guides. The use of preoperative CAD, real-time stereotactic navigation, and ultrasound-powered tools demonstrates innovation in improving surgical precision. Some concerns:

  1. The author's article exhibits concerns with respect to citation practices. The background section, in particular, demonstrates an overabundance of citations. Conversely, within the main body of the text, there are instances where critical points, warranting appropriate citation, lack the requisite referencing.

Response: The authors appreciate your time for reviewing the manuscript. We have re-reviewed all citations and agree with this assessment. We have removed 3 citations describing duplicate findings within the introduction and added a citation within the main body of text [18] to describe the STROBE technique for the retrospective analysis. Appropriate citations have also been added to sentences with critical points within the discussion section as highlighted. Also, for introduction section, relevant citations were provided for key statements.

  1. The distinctions in color within Figure 1 require clarification and elucidation.

Response: The authors agree with this assessment. We have now addressed this.

3.The limited sample size in the experiment raises questions regarding the overall validity and significance of the research findings.

Response: The authors agree with this assessment and have now included this within the limitations section in the last paragraph of the discussion. Although this is true, the objective of this case series is to report our step-by-step technique to execute the novel surgical strategy that may allow multiple large institutions to potentially adopt this technique that will allow for future prospective comparative studies.

  1. The absence of a control group or comparative analysis makes it challenging to ascertain the specific advantages of this method over conventional approaches.

Response: The authors agree with this assessment and have now included this within the limitations section in the last paragraph of the discussion. Although this is true, as mentioned above, the objective of this case series is to report our step-by-step technique to execute the novel surgical strategy that may allow multiple large institutions to potentially adopt this technique that will allow for future prospective comparative studies.

Round 2

Reviewer 1 Report

Comments and Suggestions for Authors

Dear Authors

I remind you to check figure 3 at the bottom left there is a grammatical correction signal in red "en bloc"

Best regards

Reviewer 3 Report

Comments and Suggestions for Authors

The author has undertaken revisions to address the identified issues and recommends the submission for publication.